# Multisite Is Superior to Single-Site Intratumoral Chemotherapy to Retard the Outcomes of Pancreatic Ductal Adenocarcinoma in a Murine Model

**DOI:** 10.3390/cancers15245801

**Published:** 2023-12-11

**Authors:** Janette Lazarovits, Ron Epelbaum, Jesse Lachter, Yaron Amikam, Jacob Ben Arie

**Affiliations:** 1OnePass Medical Ltd., Katzrin 1292847, Israel; ron@epelbaum.me (R.E.); yaron.a@onepassmedical.com (Y.A.); jacob@onepassmedical.com (J.B.A.); 2Meuhedet (United) Healthcare and Elisha Hospital, Haifa 3463626, Israel; jesse.lachter@gmail.com

**Keywords:** EUS-FNA, drug delivery, LAPC, stroma

## Abstract

**Simple Summary:**

The intrinsic architecture of pancreatic cancer is limiting the efficient dispersion of systemically administered drugs. The research in a mouse model presented herein aims to demonstrate that a multifocal intratumoral drug injection for wider dispersion combined with systemic therapy offers a significant clinical advantage compared to a similar treatment at the same local dosage at a single focus. The encouraging findings obtained may be translated into clinical practice to enhance the quality of life and life expectancy of pancreatic cancer patients.

**Abstract:**

Introduction: Locally advanced unresectable pancreatic cancer (LAPC) has a dismal prognosis, with intratumoral therapies showing limited benefits. We assume that the dense stroma within these tumors hampers drug dispersion. Aim: This study explores the efficacy of multisite intratumoral injections in improving a drug’s distribution while minimizing its side effects. Methods and Results: In mice with orthotopic LAPC tumors, weekly intratumoral injections of oxaliplatin at four separate sites reduced the tumor growth by 46% compared with saline (*p* < 0.003). Oxaliplatin exhibited the greatest impact on the tumor microenvironment relative to gemcitabine, Abraxane, or their combination, with increased necrosis, apoptosis, fibroblasts, inflammation, and infiltrating lymphocytes (*p* < 0.008). When combined with intravenous FOLFIRINOX (FFX), multisite intratumoral oxaliplatin reduced the tumor weight by 35% compared with single-site injection (*p* = 0.007). No additional visible toxicity was observed even at a 10-fold occurrence of intratumoral treatment. This co-modality treatment significantly improved survival compared with other groups (*p* = 0.007). Conclusions: Multisite intratumoral therapy in tandem with systemic treatment holds promise for reducing the tumor size and enhancing the overall survival in LAPC.

## 1. Introduction

The median survival in patients with locally advanced pancreatic cancer (LAPC) is typically 9–12 months [1,2]. The impaired response and resistance to current treatments are attributed, in part, to the tumor microenvironment barrier with a limited blood supply, which significantly hinders the effectiveness of systemically administered therapeutics [3,4,5]. Previous clinical trials evaluating the local delivery of chemotherapy using fine-needle endoscopic ultrasonography (FN-EUS) for LAPC have demonstrated little or no clinical improvement [6,7]. This is possibly due to the limited dissemination of the therapies injected only into a small portion of the tumor stroma, not impacting the entire tumor. We hypothesized that by spreading chemotherapies to multiple locations deep inside the solid tumors, we may overcome the barrier of desmoplastic stroma, thereby improving clinical outcomes by combining the multisite intratumoral delivery of drugs with the conventional, systemic, and potent “gold standard” treatment. 

To explore and support this approach, preclinical experiments were conducted in nude mice models of human pancreatic tumors. First, we selected the chemotherapy that has an intratumoral advantage on both the tumor growth and its microenvironment. Subsequently, we compared one intratumoral injection site versus four intratumoral injection sites at short intervals in conjunction with the systemic therapy. 

## 2. Materials and Methods

Animal studies were conducted under an ethical protocol (IL-SIA-21-6-25 approval of “The Israel Board for Animal Experiments”, in compliance with “The Israel Animal Welfare Act” and the ethics committee). 

### 2.1. PANC1 Orthotopic Pancreatic Tumor Mouse Model 

The PANC1 orthotopic pancreatic tumor mouse model, based on the method described by Qiu and Su [8], was used with some modifications, as detailed in Figure 1. At day 20 post-cell inoculation, when tumors had reached a median size of 50 mm^3^ (range 40–60 mm^3^), the mice bearing tumors were randomly divided into the various study groups.

### 2.2. Drugs and Drugs Administration 

For the systemic treatment, animals were injected through the tail vein with a mixture of a half dose of combination chemotherapy FFX: 50 mg/kg Ca + 2 folinate, 2.5 mg/kg oxaliplatin, 25 mg/kg irinotecan, and 25 mg/kg 5-fluorouracil [9]. It was diluted at a 1:100 ratio and administered either alone or in combination with intratumoral treatment. 

Intratumoral injections were performed using fine needles (Hamilton-35 g) and administered with the following total doses/treatments of chemotherapeutic agents: 7.5 mg gemcitabine [10] (EBEWE Pharma, Unterach am Attersee, Austria); 2.5 mg “Abraxane” (paclitaxel protein-bound particles for injectable suspension) [11] (Celgene, Europe B.V, Utrecht, Netherlands); 250–2500 ng oxaliplatin [12] (TEVA, Tel Aviv, Israel). For the control group or drug dilutions, a saline solution (TEVA, Tel Aviv, Israel) was used as the vehicle. The entire dose was diluted in 40 μL. In the case of a single-site injection, the entire volume was injected at once, while for four sites, the dose was divided, and 10 μL/site was injected at short intervals. The injection sites were spaced evenly, depending on the size of the tumor. 

Intratumoral, systemic, or combined treatments were administered 3–4 times weekly. Potential additional toxicity was assessed through daily observation, specifically through the monitoring of common symptoms such as changes in movement, the occurrence of diarrhea, and signs of hemorrhage, while also conducting biweekly weight measurements. In cases where a reduction of more than 20% in body weight was observed, humane euthanasia was performed on the animal.

### 2.3. In Vivo Experiments

A series of four in vivo studies were conducted to investigate the efficacy of various chemotherapies in affecting tumors. The first study, designated **s-21-271**, aimed to identify the optimal local antitumor treatment by administering different conventional chemotherapies (gemcitabine, Abraxane, and oxaliplatin) as a single agent or combination of two agents directly into the tumor. Reduction in tumor growth was compared while assessing the impact on the various components of the stromal environment. A total of eight animals were randomly assigned and subjected to weekly treatments over a period of four weeks at four distinct sites per treatment. The experimental design is visually represented in Figure 1A.

Tumor development and the initiation of treatment occurred three weeks after cell inoculation, as illustrated in Figure 1A. Subsequently, the mice harboring tumors were randomly allocated into distinct study groups, with each group comprising 10 individuals (*N* = 10). Intravenous treatments were administered on a weekly basis until the point of mortality, while intratumoral treatments were injected only four times, once a week. The animals were subjected to daily inspection of commonly observed symptoms, including fatigue, weight loss, and diarrhea. Furthermore, we conducted biweekly weight measurements for the animals until the time of their demise.

An incision of 1–2 cm in diameter was made to expose the left abdominal flank, and 1 × 10^5^ cells of PANC1 cells (ATCC) were slowly injected at P7-8 using a 27G needle. At day 20 post-cell inoculation, when tumors had reached a median size of 50 mm^3^ (range 40–60 mm^3^), the study drugs, as a single agent or combination of drugs, were injected intratumorally at the exponential growth stage. The animals were sacrificed seven days after the treatment ended. The mice bearing tumors were randomly divided into the various study groups (*N* = 5–8).

The objective of the second experiment, labeled as s-21-371, was to compare the efficacy of local administration of oxaliplatin (250 ng/treatment) at a single site (L1O-IV) versus at four sites (L4O-IV), in combination with the standard systemic care regimen (FFX). The experimental design encompassed three distinct control arms: a saline (control) group, a cohort subjected to local treatment exclusively at four designated sites (L4O), and a group subjected to systemic administration of FFX via intravenous means (IV).Each group comprised eight mice, and the experimental protocol, as above, was carried out three times only.

A third s-22-357 “dose-response experiment”, examined the effects of four different total doses of oxaliplatin (0, 250, 750, and 2500 ng/treatment) administered intratumorally at four sites in combination with systemic treatment. The primary focus was on examining the improvement in clinical outcomes without concomitant increases in toxicity. The control group received intravenous FFX (IV). Each group consisted of five mice.

The aim in the fourth survival experiment (s-23-084) was to assess the extension of animal survival between two treatment regimens: intratumoral therapy administered at four sites (4S + IV) and intratumoral therapy at a single site (SS + IV), both in conjunction with systemic treatment. Two control groups were used: one receiving saline only (control) and the other receiving systemic treatment only (IV). Systemic treatment was administered once a week until the animals died. The study design is illustrated in Figure 1B.

### 2.4. Efficacy Measurements and Analysis

The evaluation of efficacy throughout the entire study relied on the assessment of tumor volume, which was measured using electronic calipers weekly. Upon sacrifice, the tumors were excised, weighed, photographed, and preserved in formaldehyde for further analysis.

The tumors underwent a semiquantitative histopathological examination involving the analysis of slides containing sections stained with three distinct dyes: hematoxylin–eosin (H&E), Sirius red (SR, ABCAM cat# ab246832, Cambridge, UK), and Masson’s trichrome (MT, Scytek cat#TRM-500, West Logan, UT, USA). Eight common pathological parameters were evaluated during this analysis, which encompassed inflammation, lymphocyte infiltration, mitotic index, tumor necrosis, tumor matrix, nuclear pleomorphism, presence of multinucleated giant cells, and fibrosis. 

Furthermore, immunohistochemistry (IHC) techniques were employed to assess the expression levels of specific biomarkers: Ki67 staining (ABCAM cat#ab16667, Cambridge, UK) as a marker of proliferation, TUNEL assay (S7100 manual kit by Merck Millipore, Beijing, China) to measure apoptosis, and SMA staining (Novusbio cat# NBP2-33006, Centennial, CO, USA) to determine fibroblast count.

Pictures were taken using an Olympus microscope (BX60, serial NO. 7D04032) equipped with a microscope camera (Olympus DP73, serial NO. OH05504) at objective magnifications of ×1.25 and ×10. The semiquantitative scoring results are presented in Table 1 only for the parameters that showed significant differences. 

One pathologist analyzed ten nonoverlapping fields of the tumor in each section at objective magnifications as outlined in the table for each parameter. Semiquantitative analysis was used to present the pathological changes. Table 1 presents the scoring method for only the parameters that showed statistically significant differences, as depicted in the results.

### 2.5. Statistical Methods

For comparing tumor weights, two-tailed *t*-tests for independent samples were utilized, whereas the one-way ANOVA was employed when there were multiple comparison groups. A general linear model with repeated measures was applied, considering the Bonferroni correction for multiple comparisons to compare the variations in tumor volumes across the study groups over four or five consecutive time points. The semiquantitative method was used to analyze various parameters among the study groups, utilizing a nonparametric Kruskal–Wallis one-way ANOVA with Bonferroni correction as deemed appropriate. Statistical significance was established at a *p*-value < 0.05.

## 3. Results

### 3.1. Study S-21-27

Figure 2A shows all excised tumors, indicating a significant decrease in the average tumor weight (±STD) and a *p*-value of ≤0.003 when comparing the control group to each treatment group. The growth rate slope of the control group was calculated as 49.0, and following treatment, it decreased to a range of 12.4–20.1, demonstrating statistically significant inhibition of tumor growth (*p* < 0.001 in Figure 2B) in the five time points. Among the treatment groups, those administered with gemcitabine alone or in combination with oxaliplatin exhibited the most pronounced reduction in both the tumor weight and growth rate. It should be noted that in all experiments, there is a high correlation between the weight of the tumors after excision and the volume measured for each animal at the end.

The effects of various chemotherapy agents, specifically saline (control), gemcitabine (7.5 mg/treatment), Abraxane (2.5 mg/treatment), and oxaliplatin (250 ng/treatment) as single agents or in combination, on the size and tumor growth rate in an orthotopic pancreatic tumor mouse model are presented in Figure 2A and Figure 2B, respectively.

The illustration of the dissected tumor tissues at the end of the experiment is presented in Figure 2A. The tumors were photographed in groups according to the treatments indicated on top of each group. The bar added to the photo enabled us to assess the differences in the size of the tumors in each animal and between the groups. The average in tumor’s weight of each group at sacrifice was compared to the control and treatment group using a “two-tailed *t*-test for independent samples or with one-way ANOVA”, and a *p*-value of 0.003 or lower was obtained.

The average tumor volumes with time from the control and treatment groups were plotted. The dimension of the tumor volume was measured once a week using a digital caliper on the day of treatment, and at sacrifice. The tumor volume was calculated as L × W2/2 (L  = length, in millimeters; W  =  width, in millimeters). For statistical analyses, a “General linear model repeated measures with Bonferroni correction for multiple comparisons” was used. A *p*-value < 0.001 was obtained when comparing the control group to the five consecutive time points of all study groups.

Although the oxaliplatin treatment alone (250 ng/treatment) did not exhibit the highest inhibition, it induced the most notable histological changes within the tumor compared to the other tested therapies. Table 2 presents the results of the semiquantitative evaluation, revealing significant scoring differences compared to the control group (2.25 vs. 1.13 for inflammation, 1.88 vs. 1 for lymphocyte infiltration, and 3.88 vs. 2.75 for necrosis). The *p*-values for the three parameters were ≤0.005. The existence of inflammation and infiltrating lymphocytes across all groups had significant *p*-values of 0.003 and 0.015, respectively, but not in the scoring of necrosis between the groups (*p* = 0.12). Representative histological images of the pancreatic tumors treated with saline and oxaliplatin are shown in Figure 3A.

Immunostaining with smooth muscle antibodies (SMAs) for fibroblasts and the TUNEL test for apoptosis yielded promising results. The oxaliplatin treatment groups exhibited a significant reduction in fibroblast scores (1.33 vs. 3) and an increase in apoptosis (4 vs. 1.67) compared to the control group. Figure 3B displays representative histological photographs of the SMA and TUNEL tests.

Throughout the experiments, the additional treatments’ potential adverse effects were monitored. The body weight was not significantly changed, and no unacceptable toxicity or adverse events were observed. Only small tumors had developed at the point of cell inoculation.

A histopathological analysis of the tumors treated intratumorally at four sites with various drugs was performed. Sections of the tumors were stained with hematoxylin–eosin (H and E), Sirius red (SR), and Masson’s trichrome (MT) dyes. Eight common pathological parameters were semi-quantitatively analyzed. The results of the semiquantitative analysis of only inflammation, lymphocyte infiltration, and necrosis, which shows differences between the groups, are presented in Table 2. The scoring of the other parameters, which did not change, is not shown. The “Non-parametric Kruskal Wallis one-way ANOVA with Bonferroni correction”, was used to calculate the *p*-values. The differences are considered statistically significant at a *p*-value of <0.05. *p*-values of 0.003, 0.015, and 0.12, respectively, were demonstrated across all groups for the above parameters. However, when oxaliplatin scoring was compared to the control (pairwise comparisons of treatment), the *p*-values for the three parameters were <0.0001, =0.001, and 0.005, respectively.

Hematoxylin–eosin (H and E) staining was performed, and representative sections of saline- (control) and oxaliplatin-treated tumors (*N* = 8 in each group) were selected to show the differences between the groups. The images were acquired at a 1.25× magnification for fibrosis and at 10× for inflammation. The images revealed higher necrosis (stained pink) surrounded by inflammatory cells (black and yellow arrows, respectively) in the oxaliplatin-treated group. 

The representative images at a 10× magnification of α-SMA expression (fibroblasts) and TUNEL expression (apoptosis) are shown, which are significantly different from the control. Statistical analyses were performed using the “Kruskal-Wallis test”, with *p* < 0.05 considered to be statistically significant.

### 3.2. Study S-21-371

Based on the results above, 250 ng/treatment of oxaliplatin was chosen for local injection all at one site (IV + SS) or divided into four sites (IV + 4S). Despite similar conditions, the tumor growth was faster in this experiment, resulting in the death of one animal in the control group. Consequently, the experiment was terminated one week earlier than planned.

Statistically significant differences in tumor volume were observed at four consecutive time points for all groups (Figure 4A) with a *p* ≤ 0.001. Along with our hypothesis, group IV + 4S exhibited an ~30% reduction in tumor weight (638 ± 212 vs. 989 ± 233) relative to the IV + SS group (*p* = 0.007) (Figure 4B), or *p* = 0.01 by volume (Figure 4A) at sacrifice. However, no significant difference was found between the local treatment-only group (4S) and the control group (*p* = 0.057). 

In terms of safety, one animal in the control group died before treatment initiation, but no other adverse events, safety concerns, or toxic signs were observed in the remaining animals throughout the study.

The growth of the tumors over time (in volume mm^3^) was compared between a combination of single or multisite local injections with systemic treatment. The dimension of the tumor volume was measured and calculated as described in Figure 2A, where a “general linear model repeated measures for multiple comparisons” was used, and a *p*-value < 0.001 was calculated when comparing the control group at all four consecutive time points to all the study groups. Further analysis with the Bonferroni correction revealed no statistical difference between the control and the local injection groups. In a paired *t*-test between the single site vs. multiple site injection on day 21, the calculated *p*-value is 0.01.

Relative to the control, in the “One way ANOVA test”, there is a statistically significant difference between all groups (*p*-value < 0.001) in the tumor weight at sacrifice. A subsequent comparison analysis with “*t*-test for independent samples” between the single- (IV + SS) vs. multi-site (IV + 4S) local injections of oxaliplatin, both combined with the systemic treatment, indicated the greatest difference with *p* = 0.007. A nonsignificant *p*-value (>0.05) was obtained in the subsequent tests between the group treated locally only (4S) and the control group.

### 3.3. Study S-22-357

The tumor weights at the time of sacrifice exhibited a dose-dependent relationship in the treated groups, as evidenced by the following values: 1642 ± 124 mg, 1385 ± 55 mg, and 1018 ± 47 mg (*p* < 0.001). Notably, the 4S-250 group displayed minimal effects (2429 ± 123 mg) when compared to the control group (2649 ± 220 mg), albeit without reaching statistical significance despite observing significant effects in previous experiments. A statistical analysis was performed on the highest dose group (IV + 4S-2500) in comparison to each treatment group, and the corresponding results are presented in Figure 5. Only the mid-dose of 750 ng/injection did not exhibit statistical significance (*p* > 0.05).

Consistent with prior experiments, no additional adverse events, weight loss, or signs of toxicity were observed, even with a tenfold escalation in the local dose [13].

A fixed dose of systemic FFX combined with one of the three doses of intratumoral oxaliplatin divided into four sites: 250, 750, and 2500 ng/treatment in the groups’ IV + 4S-250, IV + 4S-750, and IV + 4S-2500, respectively. The control groups are saline (control), local injection alone (4S-250), and systemic treatment only (IV). Relative to the control, in “One way ANOVA”, a statistically significant difference with *p* < 0.001 in the dose-dependent inhibition of tumor weight at sacrifice was observed in the treated groups. A subsequent analysis with “Post Hoc tests with Bonferroni correction” relative to the high dose group (IV + 4S 2500) showed a significant difference in each of the various treatments, except for the mid-dose of 750 ng/injection (*p*-value > 0.05). 

### 3.4. Survival Study S-22-357

Our hypothesis posited that in addition to the impact on tumor reduction on the 21st day of treatment (*p* = 0.007), an enhanced physical local dispersion would exert a more profound overall survival rate. As shown in the Kaplan–Meier curve (Figure 6), the superiority in the median survival period for the four sites combined with intravenous treatment (4S + IV) group reached 60 days, with a substantial extension compared to the counterpart injected at a single site only (SS + IV group; 54 days), the IV group (49 days), and the control group (41 days). These differences in survival times were statistically significant, with *p*-values of 0.004, *p* < 0.0001, and *p* < 0.0001, respectively. 

The experimental protocol involved the administration of a consistent systemic FFX dosage, in conjunction with 2500 ng of oxaliplatin per treatment, which was diluted in saline. This local treatment was delivered into half of the computed tumor volume during each injection. The injection process had two variations: it was either administered at a single site (referred to as SS + IV) or evenly divided and injected into four distinct sites (referred to as IV-4S). The control groups are saline (control) and systemic treatment only (IV). 

The Kaplan–Meier survival curve demonstrated that the median survival for the IV-4S group was 60 days, whereas the SS + IV, IV-only and saline control groups had median survivals of 54, 49, and 41 days, respectively. All *p*-values were found to be ≤0.004 when comparing these groups to the control. Additionally, the comparison between the groups IV-4S and SS + IV yielded a *p*-value of 0.004.

Remarkably, the enhanced survival benefit of the local multisite treatment persisted well beyond the initial weeks of administration, even while continuing systemic therapy alone.

## 4. Discussion

The effectiveness of intravenous (IV) delivery of antineoplastic agents for treating solid tumors relies on their efficient distribution and effective targeting of malignant cells. However, various factors contribute to the limited extravasation of drugs from the tumor vasculature and their diffusion within the tumor, potentially impacting treatment outcomes [3,14,15]. These factors include irregular blood flow, high interstitial fluid pressure [16,17], and others, which can lead to drug resistance [18]. The efforts to enhance drug delivery beyond intravenous therapies have focused on strategies to overcome the physical barriers, one of which was the direct injection of drugs into the tumor. Originally designed for aiding the diagnosis and staging of pancreatic cancer, endoscopic ultrasound-guided fine-needle aspiration/biopsy (EUS-FNA/B) has been used as a tool for the local delivery of the therapy in pancreatic cancer [7,19,20]. However, no significant improvements in the outcome were demonstrated when various different conventional and innovative treatment modalities were injected intratumorally at one site [21].

The application of EUS-FNA with a single needle in the treatment of LAPC encounters various constraints. These limitations arise from the high density of the tumor, the difficulties in maneuvering the needle tip within the tumor, and the volume of the injected substance. In a cohort of patients, the intratumoral administration of gemcitabine by a single needle, in conjunction with conventional multimodality therapy, exhibited a strong association between the median overall survival and the pattern as well as the extent of intratumoral dissemination [22]. This emphasizes the need for advanced technologies that can improve the dispersion of the drug, consequently leading to enhanced long-term treatment efficacy.

The primary aim of this study was to investigate whether the broader local dispersion of a potent cytotoxic drug, which may also possess potential anti-stromal effects, could enhance the clinical outcomes of systemic treatment. To this end, we used four fine needles (35 g) for the multisite injection during a single treatment in nude mice models of human orthotopic intrapancreatic cancer, characterized by a dense tumor microenvironment [8,23]. In the initial experiment, we compared the effect of several potent drugs and demonstrated both the superior cytotoxic properties and the anti-stromal effects of oxaliplatin. Oxaliplatin induces alterations within the stromal microenvironment which inhibit tumor growth. These modifications encompass an elevation in the inflammatory response, augmentation in the infiltrating lymphocyte population, induction of apoptosis, stimulation of fibrotic processes, and reduction in the number of fibroblasts. 

Fibroblasts play a vital role in the secretion of macromolecules, which contribute to the formation of a dense microenvironment [24,25]. Stromal fibroblasts also aid in cancer cell survival, and killing them can disrupt the supportive environment [26]. Together, with the enhancement in the immune response, stromal fibroblasts hold promise in improving the effectiveness of cancer therapies and patient outcomes [23,27,28].

In our subsequent experiment, we sought to ascertain the impact of an intratumoral distribution of oxaliplatin, administered via a single-site injection versus a more widespread application across four distinct sites, in enhancing the antitumoral effects of intravenously administered FFX. Our findings revealed a substantial enhancement in therapeutic efficacy when oxaliplatin was distributed across multiple sites, resulting in a remarkable 35% reduction in both the tumor size and growth rate (*p* = 0.007). This outcome underscores the potential of optimizing intratumoral drug distribution to partially surmount the stromal barrier and achieve noteworthy therapeutic outcomes. Importantly, this effect displayed a dose-dependent relationship (*p* < 0.05).

In addition, despite oxaliplatin’s well-documented toxicity profile, no indications of additional toxicity were observed, even when the local dose was increased tenfold. Furthermore, as demonstrated in the fourth experiment, animals treated with the high dose at four distinct sites and IV treatment experienced a significant extension in their survival, highlighting the promising therapeutic potential of this approach.

Noteworthy is the fact that due to its distinctive biodistribution [29], the dose intensity of oxaliplatin administered intravenously within the FFX protocol did not correlate with progression-free survival (PFS) in patients with resected PDAC. Therefore, a pharmacokinetically-guided dose adjustment of this drug to improve survival outcomes in these patients was not considered [30]. Our study involved the direct intratumoral injection and distribution of oxaliplatin, thus, surpassing the constraints imposed by its distinctive biodistribution. 

Our results supported the hypothesis that the multisite intratumoral drug delivery strategy in LAPC improves the systemic treatment’s efficacy and can provide a basis for future clinical trials to enhance the overall outcomes of systemic treatments. However, in this study, we were limited in the number of treatments because of the need to maintain an open abdominal cavity for each of the animals during the study, which increased the risk of infections. We anticipate that this limitation will be less significant in human patients. Despite the encouraging results, translating the procedure to a clinical setting presents challenges. One major concern is ensuring controlled injection in humans. Therefore, it would be essential to develop a method for the uniform dispersion of injected drugs intratumorally, while preventing leakage into neighboring tissues. Lastly, this model is inadequate for demonstrating the tumor’s transition from nonresectable to resectable, which is a valuable parameter for improving treatment efficacy in the LAPC patient group where there is still potential for improvement.

## 5. Conclusions

In this study, the utilization of a generic antineoplastic agent in a PDAC model revealed the superiority of the spread of the drug injected through a multisite intratumoral drug delivery strategy over single-site local chemotherapy. By overcoming the intratumor barrier and spreading the drugs administered locally, this approach enhanced the clinical outcomes of systemic drug delivery in inhibiting tumor growth, with no visible side-effects. Oxaliplatin was employed in this study to demonstrate its feasibility as a treatment option. The findings provide a proof-of-concept for future utilization in the local delivery of various potent and innovative drugs with diverse mechanisms of action in upcoming applications.

## Figures and Tables

**Figure 1 cancers-15-05801-f001:**
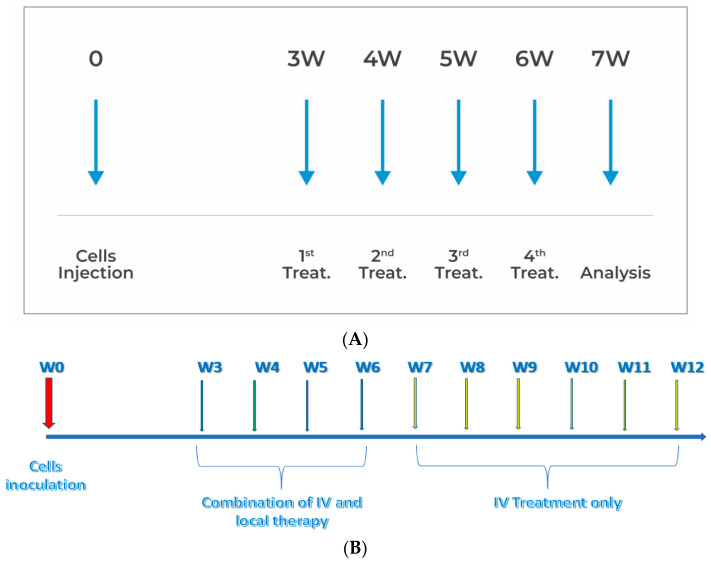
(**A**) Design of the in vivo experiments. (**B**) Survival study design.

**Figure 2 cancers-15-05801-f002:**
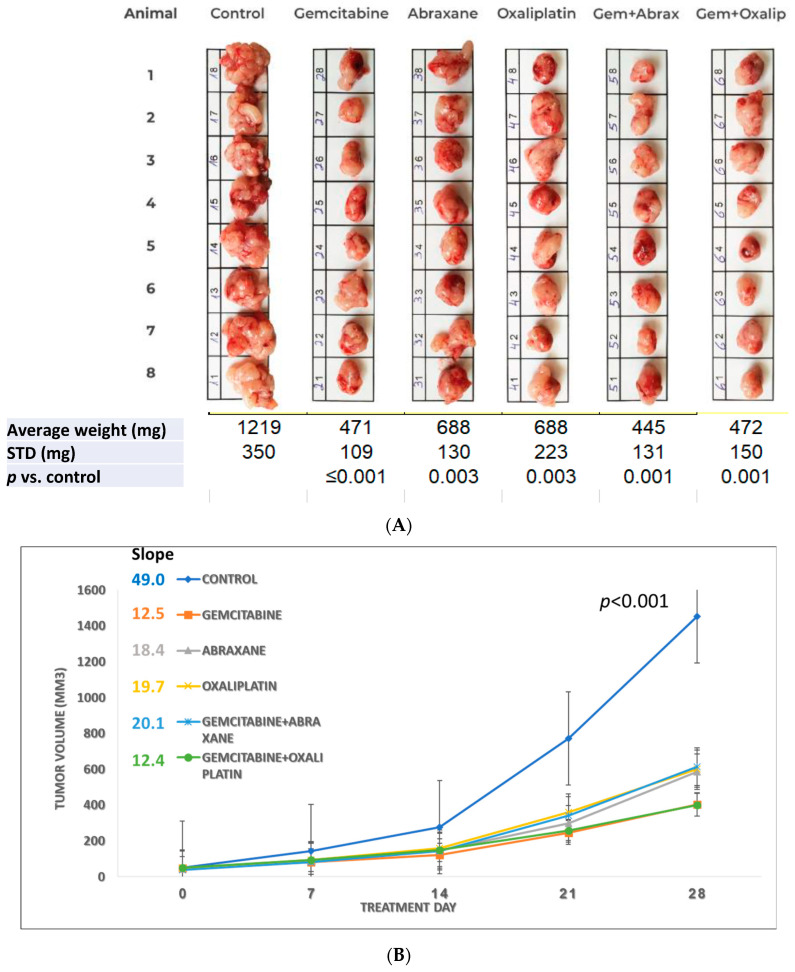
The effect of multisite intratumoral injection on the orthotopic pancreatic ductal adenocarcinoma (PDAC) model. (**A**) Images and tumor weights at sacrifice; (**B**) changes in tumor volume (mm^3^) from the beginning of treatment to the day of sacrifice.

**Figure 3 cancers-15-05801-f003:**
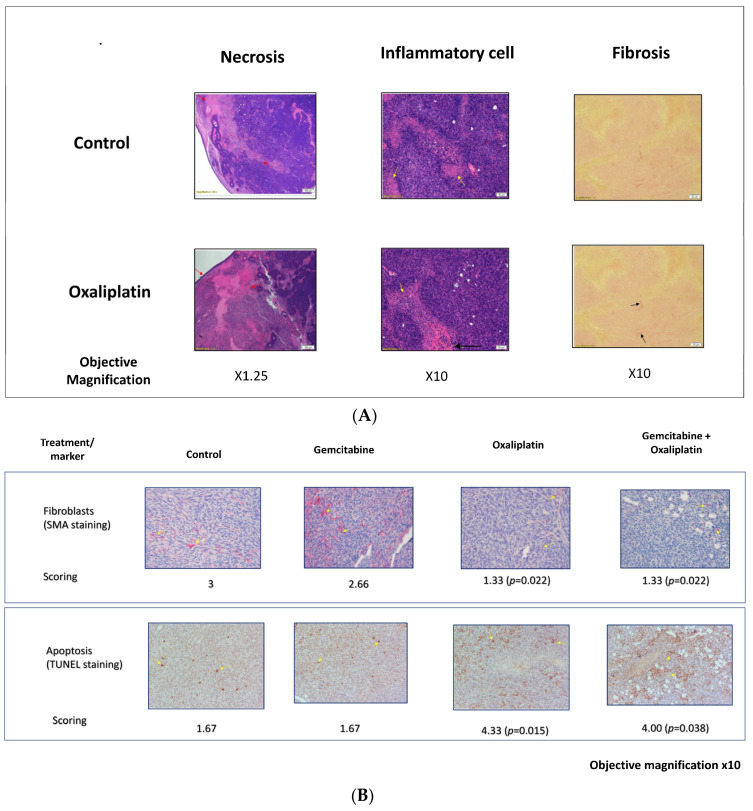
Histologic and immunohistological analysis of pancreatic tumors. (**A**) Representative sections of tissue from pancreatic tumors were subjected to histologic analysis of paraffin-embedded slides. Arrows are indicating staining locations. (**B**) Immunohistology assessment of the changes in the excised tumors for specific biomarkers.

**Figure 4 cancers-15-05801-f004:**
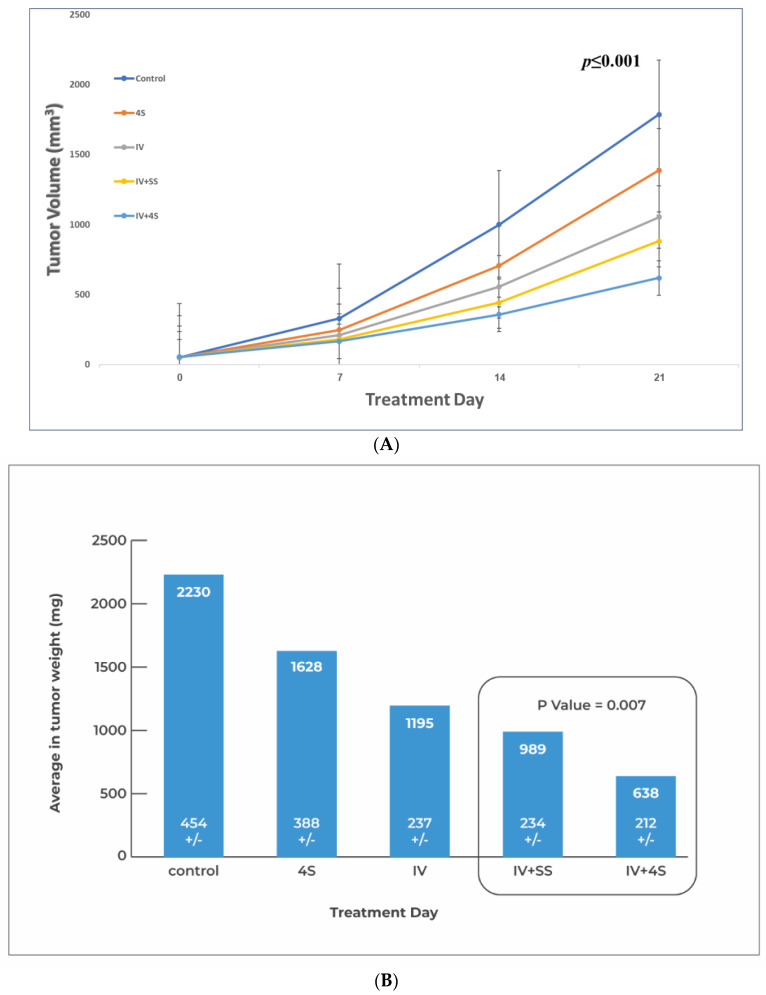
Effects of single vs. multiple intratumoral injections adjacent to IV treatment on pancreatic tumors in orthotopic mice models. Tumor formation and scheduling of treatments are described in Figure 1, with the exception that the animals received 3 treatments once a week and left for a one-week follow-up before sacrifice. For local treatment, a total of 250 ng of oxaliplatin was injected intratumorally each time all at one site or the dose was divided into four sites. FFX (a mixture as described in Section 2) was injected intravenously. The test groups received combination therapy. In the group IV + SS, all the 250 ng of oxaliplatin was injected at one site. In the group IV + 4S, the 250 ng oxaliplatin was divided and injected at 4 sites. Control groups are included: saline injection (control), only IV administration of FFX (IV group), and only intratumoral injection of 250 ng oxaliplatin at 4 sites (4S). In each group, *N* = 8. (**A**) Growth of tumors over time (in volume mm^3^). (**B**) Average tumor weight at sacrifice for each group.

**Figure 5 cancers-15-05801-f005:**
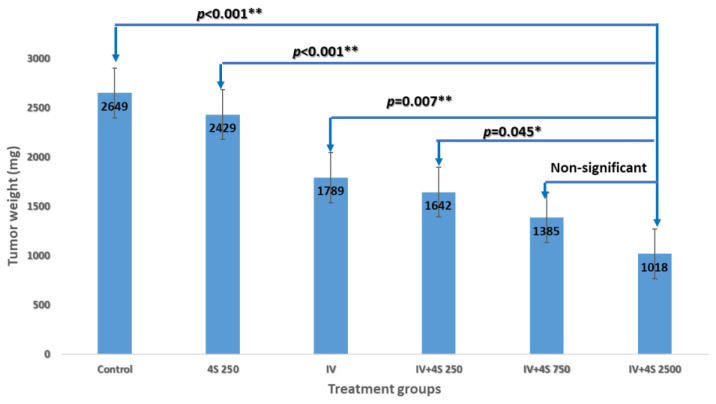
Dose response of the local treatment combined with systemic therapy.

**Figure 6 cancers-15-05801-f006:**
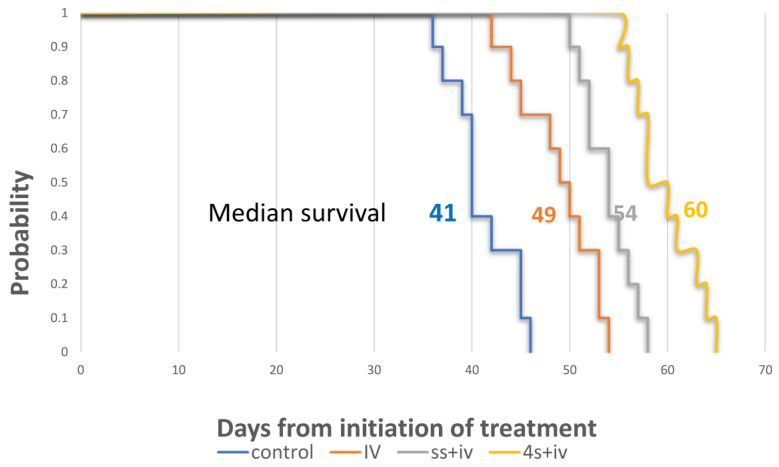
Survival analysis of combined local and systemic therapies.

**Table 1 cancers-15-05801-t001:** Semiquantitative scoring system for the presence of pathological changes.

Parameter (Field)	Magnification (X)	Score-0	Score-1	Score-2	Score-3	Score-4	Score-5
Inflammatory cells (H&E)	10	Rare or absent	<tumor cells	=tumor cells	Predominant		
Lymphocytes infiltration (H&E)	10	Absent	<5	5–20	>20		
Necrosis (H&E)	10	No sign	Small foci	<50%	50–75%	75–90%	>90%
Apoptotic cells (TUNEL)	20	No positive	<5	5–15	15–25	25–50	>50
Fibroblasts(SMA)	20	No positive	<5	5–15	15–25	25–50	>50

**Table 2 cancers-15-05801-t002:** Semiquantitative histological analysis of the tumor.

Parameter	Treatment/Number of Animal in Each Score					
Inflammatory Infiltration	Control	Gemcitabine	Abraxane	Oxaliplatin	Gemcitabine + Abraxane	Gemcitabine + Oxaliplatin
Score-1	7	3	1	0	2	0
Score-2	1	2	4	3	5	6
Score-3	0	3	3	5	1	2
Mean	1.13	2.00	2.25	2.63	1.88	2.25
*p*-value	All groups = 0.003	0.021	0.003	<0.0001	0.05	0.003
Lymphocytes infiltration						
Score-1	8	5	3	1	3	3
Score-2	0	3	5	7	5	5
Score-3	0	0	0	0	0	0
Mean	1.00	1.38	1.63	1.88	1.63	1.63
*p*-value	All groups = 0.015	0.137	0.013	0.001	0.013	0.013
Tumor necrosis						
Score-2	2	1	1	0	2	1
Score-3	6	4	2	3	2	4
Score-4	0	3	5	3	4	3
Score-5	0	0	0	2	0	0
Mean	2.75	3.25	3.50	3.88	3.25	3.25
*p*-value	All groups = 0.120	0.163	0.032	0.005	0.133	0.163

## Data Availability

The data presented in this study are available in this article.

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
