# Peer review of "Multisite Is Superior to Single-Site Intratumoral Chemotherapy to Retard the Outcomes of Pancreatic Ductal Adenocarcinoma in a Murine Model"

_cancers, 2023, doi:10.3390/cancers15245801_

Round 1

Reviewer 1 Report

Comments and Suggestions for Authors

The authors of the manuscript entitled " Multisite is Superior to single site intratumoral chemotherapy to retard the outcomes of pancreatic ductal adenocarcinoma in a murine model" by Janette Lazarovit et al have presented in this manuscript how a multisite intratumoral drug delivery strategy is better than a single site local chemotherapy. The authors claim that overcoming the intratumor barrier and spreading the drugs administered locally, the clinical outcomes of the systemic drug in inhibiting tumour growth is achieved, with no visible side effects. The authors have carried out a well designed study, however I strongly believe that the authors should mention a short paragraph on the need for prolonged action drugs and how their half life can play an important role in the drug efficacy. The authors also mention no side effects, but do not highlight what are the parameters they looked into to study the side effects considering Oxaliplatin use and liver toxicity and liver tissue morphology. While the authors looked into the general side effects such as the occurrence of diarrhea, and signs of hemorrhage a closer molecular investigation would be beneficial. Additionally, multiple sites of injections also leads to puncturing of the tissue and I am curious if the authors have considered repair and healing at the site of drug administration. Overall, I believe the authors should revisit their manuscript and incorporate key changes to the manuscript especially highlighting the need or rationale for  multiple injection sites rather than efficient drugs and targeted drug delivery. Further more the authors should also spend some time in working on the figure legends and provide necessary explanations. 

By overcoming the intratumor barrier and spreading the drugs administered locally, this approach enhanced the clinical outcomes of systemic drug delivery in inhibiting tumor growth, with no visible side-effects. Oxaliplatin was employed in this study to demonstrate its feasibility as a treatment option. The findings provide a proof of concept for the future utilization of local delivery of various potent and innovative drugs with diverse mechanisms of action in upcoming applications. Author Contributions: Funding: Institutional Review Board Statement: Informed Consent Statement: Data Availability Statement: Conflicts of Interest

Author Response

Answers to 1st reviewer

  • The authors should mention a short paragraph on the need for prolonged action drugs and

Answer:

In the research presented herein, we do not assert an extension of the drug's duration of activity. Our primary assertion is that we have successfully improved the drug's dispersion immediately after injection, allowing for broader exposure of more tumor regions to the medication.

Current injections into pancreatic tumors with one needle have very limited access avenues for brachytherapy.  In our work, we clearly show that local injection is not enough into this dense tumor mass. There is also a great need for multiple access sites, and better application of the drug to the tumors to disperse the drug injected directly into the tumor. Applying to a small part of a tumor would seem less effective than to more or all of the tumors being targeted.

  • How their half life can play an important role in the drug efficacy.

Answer:

Given that the chemotherapy is administered directly into the tumor site, circumventing the need to traverse the bloodstream as in intravenous delivery, it results in immediate drug exposure, rendering the drug's half-life less significant in the context of the proposed treatment protocol.

  • The authors also mention no side effects, but do not highlight what are the parameters they looked into to study the side effects considering oxaliplatin use and liver toxicity and liver tissue morphology. While the authors looked into the general side effects such as the occurrence of diarrhea, and signs of hemorrhage

Answer:

In general, systemic chemotherapy can result in a range of side effects, including liver toxicity. The proposed treatment protocol entails augmenting the systemic therapy, which includes oxaliplatin. Our primary inquiry pertains to whether this augmentation will yield additional effects beyond the anticipated ones. We anticipate the possibility of supplementary side effects if there is any inadvertent leakage beyond the local injection site.

In our assessment of potential side effects, we have referred to established benchmarks based on animal experiments of a similar model, which involve close monitoring for substantial changes such as in weight (exceeding 20%) or alterations in behavioral parameters. The known side effects associated with systemic treatment will persist in this scenario.

In the text of the manuscript, I added "additional side effects/toxicity" to what is expected from the systemic treatment with FFX".

  • A closer molecular investigation would be beneficial.

Answer:

In this experiment, we used a well-known chemotherapy with a known mechanism of action to evaluate the effectiveness of a therapeutic protocol, rather than investigating the specific mechanism of action of the intratumorally injected drug itself. The primary objective of the experiments conducted was to demonstrate that optimizing the dispersion of chemotherapy to a tumor characterized by its dense stroma with limited diffusion, to more regions would confer a notable advantage.

  • Additionally, multiple sites of injections also leads to puncturing of the tissue and I am curious if the authors have considered repair and healing at the site of drug administration.

Answer:

Yes. As outlined in the methods chapter, for the injections, the abdominal cavity was sealed with a clip, enabling the tumor's exposure for subsequent injections. Consequently, the injection site was visually apparent as healed. It is noteworthy that we used a 35-gauge thin needle for the injections.

  • Overall, I believe the authors should revisit their manuscript and incorporate key changes to the manuscript especially highlighting the need or rationale for  multiple injection sites rather than efficient drugs and targeted drug delivery.

Answer:

I would like to draw your attention to the sentences already written in the introductory and discussion chapters of the manuscript.

  1. Impaired response and resistance to current treatments are attributed, in part, to the tumor microenvironment barrier with limited blood supply significantly hindering the effectiveness of systemically administered therapeutics
  2. … delivery of chemotherapy using fine-needle endoscopic ultrasonography (FN-EUS) for LAPC has demonstrated little or no clinical improvement. Possibly due to the limited dissemination of the therapies injected only into a small portion of the tumor stroma
  3. We hypothesized that by spreading chemotherapies to multiple locations deep inside solid tumors, we may overcome the barrier of desmoplastic stroma….
  4. The effectiveness of intravenous (iv) delivery of antineoplastic agents for treating solid tumors relies on their efficient distribution and effective targeting of malignant cells. However, various factors contribute to the limited extravasation of drugs from the tumor vasculature and their diffusion within the tumor, potentially impacting treatment outcomes[10,11]. These factors include irregular blood flow, high interstitial fluid pressure, and others, which can lead to drug resistance[12]. Efforts to enhance drug delivery beyond intravenous therapies have focused on strategies to overcome physical barriers, one of which was a direct injection of drugs into the tumor.
  5. The application of EUS-FNA with a single needle in the treatment….. Limitations arise from the high density of the tumor, the difficulties in maneuvering the needle tip within the tumor, and the volume of the injected substance.
  • Further more the authors should also spend some time in working on the figure legends and provide necessary explanations. 

Answer:

I want to ensure that you have observed that the legend description for each graph is presented in italics beneath each respective graph within the manuscript's main content. These descriptions consistently outline the experimental procedures, ensuring accurate replication, and provide key findings from each treatment group, complete with their corresponding labels. 

Reviewer 2 Report

Comments and Suggestions for Authors

In the manuscript, authors have compared the effect of intra-tumoral injection of the drugs at multiple sites vs single site. they report that injecting the drug at multiple sites compared to single site is advantageous.  By injecting drug at multiple sites, the volume of the injection is being reduced compared to when it is injected at single site. It is not clear if area of spread of the drug and thereby exposure of cancer cell is being affected by changing the protocol. Additional experiment by using a trackable substitute of drug and comparing the two protocols can provide this information. When comparing administration at single site vs. multiple sites the reported statistical comparison was used on groups, which have been simultaneously given intravenous administration of the drug. Statistical analysis of groups which were administered drug at single site vs. multiple sies should be provided. Many limitations of the study have been discussed but in clinic will the recommended drug administration protocol have significant advantage keeping in view marginal survival advantages in the study?

Reviewer 3 Report

Comments and Suggestions for Authors

The authors present evidence that intrtumoral injection is beneficial for chemotherapy even in locally advanced tumors. However the study design does not include a control group where the chemotherapy was applied only as i.v. or i.p. treatment. Therefore the added impact of the intratumoral injection cannot be assessed correctly.

Author Response

  • … the study design does not include a control group where the chemotherapy was applied only as i.v. Or i.p. Treatment. Therefore the added impact of the intratumoral injection cannot be assessed correctly.

Answer:

The primary focus of this study was to investigate the comparative impacts of multi-site versus single-site intratumoral chemotherapy delivery when used as a complement to standard care therapy (SCT). In our findings, we demonstrated the enhanced efficacy of administering drugs through a multi-site intratumoral drug delivery approach, highlighting its superiority over single-site localized chemotherapy.

The objectives of each one of the experiments included in this manuscript are described in the method section.

In the experimental design, a control group was included, which received intravenous treatment with SCT. Graphs 4a, 4b, 5, and 6 from experiments study s-21-371, study s-22-357, and study s-22-358, respectively, depict the results of the control group exclusively treated via intravenous administration of SCT.

It's worth noting that SCT consists of a combination of four drugs, with oxaliplatin being one of them. The local injection dosage of oxaliplatin was determined, considering the proportion of the dose relative to the body weight.

Round 2

Reviewer 1 Report

Comments and Suggestions for Authors

I thank the authors for clarifying the questions raised.